# Approximation Preserving Coresets

**Milind Prabhu** [1]  **Chris Schwiegelshohn** [2]  **Sudarshan Shyam** [2]

## Abstract

Clustering in a big data setting is an intensively studied problem, with coresets emerging as one of the important paradigms in this line of work. Given a cost function $\text{cost}(P, S)$ mapping input points $P$ and a solution $S$ to an objective value, a coreset is a typically weighted sketch $\Omega \subseteq P$ such that $\text{cost}(\Omega, S) \approx \text{cost}(P, S)$. In practice, coreset sizes much smaller than those suggested by theoretical guarantees are often found to be sufficient. In this paper, we offer an explanation for this phenomenon. Smaller coreset sizes suffice if we only wish to preserve the costs of *good* solutions, i.e., solutions with low cost. We define and devise *approximation-preserving coresets*, which provide a weaker guarantee than strong coresets, which apply to all solutions, while providing stronger guarantees than weak coresets, which apply only to the optimum solution. We complement this result by showing that even a very small distortion in the approximation factor cannot admit coresets of this size.

## 1. Introduction

Clustering is a popular tool to partition and preprocess data in a machine learning pipeline, in particular when the data set is very large. To manage such data sets, a number of techniques have arisen to make clustering algorithms more scalable. Arguably, the most important technique in this line of work is *coresets*, see (Feldman, 2020; Munteanu & Schwiegelshohn, 2018), which downsample the data to obtain a summary that behaves similarly to the full dataset with respect to the clustering cost. As a rule of thumb, coresets can often be computed either in linear time (Cohen et al., 2017; Draganov et al., 2024), or at least very efficiently compared to any optimization algorithm. Thus, instead of running an algorithm on the full input, one can first construct

a much smaller coreset and then run the algorithm on this summary, leading to significant savings in time and memory.

Consider the Euclidean $k$-means problem. Here, given $P \subseteq \mathbb{R}^d$, the goal is to find $k$ centers $S$ minimizing

$$\text{cost}(P, S) = \sum_{p \in P} \min_{s \in S} \|p - s\|_2^2.$$

A *strong coreset* $\Omega$ is a (weighted) subset of $P$ such that, for every $k$-center set $S$,

$$|\text{cost}(P, S) - \text{cost}(\Omega, S)| \leq \varepsilon \cdot \text{cost}(P, S). \tag{1}$$

For Euclidean $k$-means, the current state-of-the-art *strong coreset* bounds (Cohen-Addad et al., 2021a; 2022b; Huang et al., 2024) give $\tilde{O}\left(k\varepsilon^{-2} \min\{\sqrt{k}, \varepsilon^{-2}\}\right)$ points,[1] and this bound is known to be optimal in the worst case (Huang et al., 2024). At the same time, it often appears overly pessimistic in practice: empirical studies frequently report that sampling on the order of $O(k)$ points already preserves costs quite accurately (Schwiegelshohn & Sheikh-Omar, 2022). Several works have proposed explanations for this gap by identifying additional structure under which much smaller summaries suffice. For instance, *lightweight* coresets (Bachem et al., 2018) are smaller but measure error relative to the 1-means cost, making them most effective when the data is not well clusterable; when the data is highly clusterable, substantially smaller coresets are also possible (Bansal et al., 2024).

In this paper, we offer another perspective on why worst-case strong coreset bounds can be pessimistic in practice: coresets are rarely queried on arbitrary solutions. Instead, they are used as preprocessing; one constructs a coreset and then runs a clustering algorithm on the summary, evaluating only the solutions that the algorithm produces. In particular, many pipelines use approximation algorithms that return a solution whose cost is guaranteed to be no worse than a factor $\alpha$ times the cost of an optimal solution. Examples include $k$-means++ seeding (an $\alpha \in O(\log k)$) (Arthur & Vassilvitskii, 2007) or local search ($\alpha \in O(1)$) (Arya et al., 2001; Kanungo et al., 2002; Cohen-Addad et al., 2022). Since these algorithms can still be expensive on large datasets, it is especially valuable to reduce the input as much as possible before running them.

[1]University of Michigan, USA [2]Aarhus University, Denmark. Correspondence to: Chris Schwiegelshohn <schwiegelshohn@cs.au.dk>.

*Proceedings of the 43$^{rd}$ International Conference on Machine Learning*, Seoul, South Korea. PMLR 306, 2026. Copyright 2026 by the author(s).

---

[1]We use $\tilde{O}(x)$ to suppress poly log factors in $x$.

A strong coreset guarantee (Equation 1) is sufficient to preserve the performance of any approximation algorithm, since it preserves the cost of *every* solution. The key question is whether this uniform guarantee is actually necessary. In particular:

> Are smaller summaries possible if we only require a sketch to *preserve the approximation guarantees* of clustering algorithms?

### 1.1. Our Contributions

We answer the above question in the affirmative. We introduce the notion of *approximation preserving coresets* (APCs), which guarantee that a solution computed on the summary can be efficiently post-processed into a good approximate solution for the full input.

**Definition 1.1** ($\beta$-Approximation Preserving Coresets)**.** A (weighted) set $\Omega$ is called a $\beta$-approximation preserving coreset for $P$ if, given any $\alpha$-approximate $k$-means solution $S$ for $\Omega$, there exists an algorithm that uses $\Omega$ and $S$ to compute an $\alpha\beta$-approximate $k$-means solution $S'$ for $P$.

We first show that Euclidean $k$-means admits small approximation-preserving coresets.

**Theorem 1.2.** *There is an algorithm to construct a $(1 + \varepsilon)$-APC of size $\tilde{O}(k\varepsilon^{-3})$ for Euclidean $k$-means.*

For comparison, optimal strong coresets for Euclidean $k$-means have worst-case size $\tilde{O}\left(k\varepsilon^{-2} \min\{\sqrt{k}, \varepsilon^{-2}\}\right)$, illustrating that preserving only approximation guarantees can yield smaller summaries.

We next study APCs for $k$-means in arbitrary metric spaces. For an $n$-point metric, strong coresets have optimal worst-case size $\Theta(k\varepsilon^{-2}\log n)$, and more generally known strong coreset bounds typically depend on structural properties of the metric (e.g., intrinsic dimension such as doubling dimension, or restrictions to special metric families such as graph-induced metrics). In contrast, we obtain $(4+\varepsilon)$-APCs for arbitrary metrics whose size is independent of input size and metric-dependent parameters.

**Theorem 1.3.** *There is an algorithm to construct a $(4 + \varepsilon)$-APC of size $\tilde{O}(k\varepsilon^{-2})$ for $k$-means on any metric.*

Moreover, for some $n$-point metrics, any $\beta$-APC with $\beta < 4$ must have size $\Omega(\log n / \log \log n)$, showing that a factor of 4 is essentially necessary to avoid dependence on $n$.

Beyond coresets, this result also has algorithmic implications: it yields a fixed-parameter tractable (FPT) $(4 + \varepsilon)$-approximation guarantee for $k$-means in arbitrary metric spaces. Notably, this factor is close to the best known FPT approximation ratio of $(1 + \frac{8}{e} + \varepsilon) \approx 3.94$ for metric $k$-

means (Cohen-Addad et al., 2019)[2]. The running time of our algorithm is $O(nk) + (\varepsilon^{-1} \cdot \log k)^{O(k)}$.

Our algorithms retain the efficiency typically expected of coresets. The APC can be constructed in nearly linear time in the input size in Euclidean space and in optimal time in general metrics. Moreover, given an approximate solution $S$ on the coreset, the corresponding approximate solution $S'$ for the original input can be recovered efficiently. Finally, the constructions are composable: APCs built for disjoint subsets can be merged into an APC for their union, making them well suited for streaming and distributed settings.

**Sensitivity sampling.** Both APC constructions are based on *sensitivity sampling*, a simple, widely used coreset algorithm (Algorithm 1). The heart of our analysis is a new concentration bound for sensitivity sampling (Theorem 2.2), which we use as a black box in our results. We expect this theorem to be of independent interest. For example, it yields strong coresets in finite $n$ point metrics of size $O(k\varepsilon^{-2}\log n)$. This construction matches the lower bound from (Cohen-Addad et al., 2022a) and is, to the best of our knowledge, the first analysis known to be optimal; all previous analyses lost polylog factors.[3]

### 1.2. Related Work

The strong coreset guarantee as defined in Equation 1 was first proposed by (Har-Peled & Mazumdar, 2004). Over the next two decades, there has been substantial activity in this field. For Euclidean $k$-means specifically, the early algorithms (Har-Peled & Mazumdar, 2004; Frahling & Sohler, 2005; Har-Peled & Kushal, 2007) tended to use geometric techniques and typically gave coresets with an exponential dependency on $d$. The modern approach of using sampling and learning theory was initiated in the seminal work by Chen (Chen, 2009). Shortly thereafter, (Langberg & Schulman, 2010) proposed sensitivity sampling for coresets, which was further improved by (Feldman & Langberg, 2011) . Subsequently, most works tended to focus on dimension reduction (Becchetti et al., 2019; Cohen-Addad et al., 2021b; 2023; Feldman et al., 2020; Huang & Vishnoi, 2020). (Cohen-Addad et al., 2021a) discovered the group sampling algorithm, an easier to analyze if slightly slower and more cumbersome alternative to sensitivity sampling and used it to prove several optimal coreset bounds

---

[2]Their guarantee is slightly stronger in that candidate centers are not necessarily part of the input. Nevertheless, we believe that even the case where input points are always candidate centers is interesting.

[3]The magnitude of these additional log factors is often not stated. However, all previous work seems to have required at least an additional factor $\log k \log \log n$ (Feldman & Langberg, 2011; Cohen-Addad et al., 2025a), or $\log^5 \varepsilon^{-1}$ (Cohen-Addad et al., 2021a).

including the aforementioned $\tilde{O}(k\varepsilon^{-4})$ bound for $k$-means. For small $\varepsilon$, this was further improved independently by (Cohen-Addad et al., 2022b) and (Huang et al., 2024) to $\tilde{O}(k\sqrt{k}\varepsilon^{-2})$. These bounds were subsequently matched by sensitivity sampling in (Bansal et al., 2024). The first non-trivial lower bounds of the order $\Omega(k\varepsilon^{-2})$ were given by (Cohen-Addad et al., 2022a), and subsequently improved by (Huang et al., 2024) to match the state of the art upper bounds up to logarithmic factors.

Other metrics followed the same frameworks as introduced for Euclidean spaces, but required other ways to characterize the number of solutions. Examples include finite metrics (Chen, 2009; Feldman & Langberg, 2011; Cohen-Addad et al., 2021a), doubling metrics (Huang et al., 2018; Cohen-Addad et al., 2021a), graph metrics (Baker et al., 2020; Bandyapadhyay et al., 2023; Braverman et al., 2020; Cohen-Addad et al., 2025a), time series (Braverman et al., 2022; Huang et al., 2021; Cohen-Addad et al., 2025a; Conradi et al., 2023).

A moral predecessor to approximation preserving coresets is the notion captured by weak coresets. While there is no unified notion, they all require that one can use the coreset to extract a (near) optimal solution. The earliest example of such a guarantee probably dates back to the minimum enclosing ball problem, see (Badoiu & Clarkson, 2008), but several examples exist for other objectives as well, see (Carmel et al., 2025; Carmel & Krauthgamer, 2026; Woodruff & Yasuda, 2024) and references therein. For $k$-means clustering, such coresets were proposed in (Blömer et al., 2018; Feldman et al., 2007; Feldman & Langberg, 2011; Jaiswal & Kumar, 2024). Historically, these coresets were commonly used to obtain $(1 + \varepsilon)$-approximations on small summaries while bypassing the curse of dimensionality applying to strong coresets, see for instance (Badoiu & Clarkson, 2008; Feldman et al., 2007). Our guarantee is stronger in that it applies to the performance of any algorithm and not only to a typically computationally expensive algorithm used to extract near optimal solutions on weak coresets.

### 1.3. Technical Overview

We begin by recalling how strong coresets for clustering are typically analyzed, using Euclidean $k$-means as a representative example. The standard analysis therefore proceeds in two steps. First, one proves that for a *fixed* set of centers $S$, the coreset cost, $\text{cost}(\Omega, S)$, concentrates around the true cost, $\text{cost}(P, S)$, with high probability. Second, we enumerate over all possible solutions. Since this number is infinite in Euclidean space, we discretize the solution space via a net argument: one constructs a finite set of candidate solutions called *net* and argues that preserving costs for all solutions in the net suffices to preserve costs for every solution, followed by a union bound over the net.

At a high level, the coreset size is dictated by the size of this net, i.e., by the number of candidate solutions whose costs must be preserved. In particular, we show later that for any fixed family $\mathcal{S}$ of $N$ solutions, a coreset of size

$$O\big(\varepsilon^{-2}\big(k \log k + \log N\big)\big) \qquad (2)$$

suffices to preserve the costs of all solutions in $\mathcal{S}$ up to multiplicative $(1 \pm \varepsilon)$ factors.

Existing strong coreset constructions for Euclidean $k$-means rely on $\varepsilon$-nets over the space of all solutions. At the resolution needed for multiplicative cost preservation, such a net has size about $N = \exp(\tilde{\Theta}(k/\varepsilon^2))$, yielding a $\tilde{O}(k\varepsilon^{-4})$ coreset, which is known to be optimal when $k \gg \varepsilon$ (Huang et al., 2024). Importantly, strong coresets must preserve even highly pathological solutions that no reasonable algorithm would ever output. APCs exploit this slack: rather than targeting *all* $k$-tuples of centers, we only need to preserve costs for the "meaningful" solutions that can arise as outputs (or near-outputs) of approximation algorithms, thereby reducing the effective $N$ and sidestepping strong coreset barriers.

**Witness Sets.** This shifts the goal from preserving costs for *all* solutions to a more targeted question: which solutions must have their costs preserved so that any $\alpha$-approximate solution computed on $\Omega$ can be efficiently converted into an $\alpha\beta$-approximate solution for $P$?

We formalize this via *witness sets*: a finite family $\mathcal{S}$ of $k$-center sets satisfying two properties.

**(1) Preserving an optimal solution.** $\mathcal{S}$ contains an optimal solution $C^* \in \arg\min_{|C|=k} \text{cost}(P, C)$.

**(2) Mapping property.** For every solution $S$, there exists $S' \in \mathcal{S}$ such that

$$\text{cost}(\Omega, S') \le \beta \cdot \text{cost}(\Omega, S).$$

Assume $\Omega$ preserves costs for all $S \in \mathcal{S}$ up to $(1 \pm \varepsilon)$ factors. It is immediate from (1) that

$$\text{OPT}(\Omega) \ \le \ \text{cost}(\Omega, C^*) \ \approx \ \text{cost}(P, C^*) \ = \ \text{OPT}(P),$$

so the coreset optimum cannot be much larger than the true optimum. Now let $S$ be an $\alpha$-approximate solution on $\Omega$. By (2), we can round $S$ to $S' \in \mathcal{S}$ with $\text{cost}(\Omega, S') \le \beta\text{cost}(\Omega, S)$. Transferring this bound from $\Omega$ back to $P$ using cost preservation on $\mathcal{S}$ gives

$$\text{cost}(P, S') \ \approx \ \text{cost}(\Omega, S') \ \le \ \beta \, \text{cost}(\Omega, S)$$
$$\le \ \alpha\beta \, \text{OPT}(\Omega) \ \lesssim \ \alpha\beta \, \text{OPT}(P).$$

Thus, preserving costs on $\mathcal{S}$ is sufficient to convert any $\alpha$-approximate solution on $\Omega$ into an $\alpha\beta$-approximate solution for $P$, i.e., $\Omega$ is a $\beta$-APC.

Note that while it may be tempting to set $\mathcal{S}$ to be the optimal solution of $\Omega$, there are several issues with this argument. First, the mapping function may not be efficiently computable. Second, and more importantly, $\mathcal{S}$ now heavily depends on $\Omega$ and thus it is difficult to use the randomness of $\Omega$ to prove concentration.

The remaining question is therefore algorithmic and geometric: how do we efficiently construct *small* witness sets for concrete clustering objectives that satisfy the above mapping property? We answer this by exploiting structure in the solution space that allows arbitrary coreset solutions to be rounded into a much smaller family.

**Witness Sets for Euclidean Space.** We now sketch the construction of witness sets for Euclidean spaces.

The key geometric fact about Euclidean space is that the empirical mean of a small uniform sample closely approximates the true mean (Bertolotti et al., 2026; Inaba et al., 1994; Cohen-Addad et al., 2025b): for any point set $C \subset \mathbb{R}^d$ with mean $\mu(C)$, there exists a subset $T \subseteq C$ of size $O(1/\varepsilon)$ whose mean $\mu(T)$ satisfies

$$\text{cost}(C, \mu(T)) \leq (1 + \varepsilon)\text{cost}(C, \mu(C)).$$

Equivalently, every cluster has a $(1 + \varepsilon)$-approximate mean that is the average of only $O(1/\varepsilon)$ input points.

Define $\mathcal{M}$ to be the set of all means of $O(1/\varepsilon)$-subsets of the input, and let the witness family $\mathcal{S}$ consist of all $k$-tuples of centers from $\mathcal{M}$. Given an arbitrary solution $S$, form its induced clusters and replace each center by a $(1 + \varepsilon)$-approximate cluster mean from $\mathcal{M}$. This is exactly a single (Lloyd-style) *re-centering step* (Lloyd, 1982) followed by snapping each center to a structured representative. The resulting $S' \in \mathcal{S}$ satisfies $\text{cost}(\Omega, S') \leq (1+O(\varepsilon))\text{cost}(\Omega, S)$, yielding the mapping property with $\beta = 1 + O(\varepsilon)$.

Since $|\mathcal{M}| \leq n^{O(1/\varepsilon)}$, we have $|\mathcal{S}| \leq |\mathcal{M}|^k \leq n^{O(k/\varepsilon)}$ and hence $\log|\mathcal{S}| = O(k\varepsilon^{-1}\log n)$. For Euclidean spaces, due to the existence of strong $\varepsilon$-coresets of size $\text{poly}(k/\varepsilon)$ we can further assume that $n = \text{poly}(k/\varepsilon)$ which means that $\log|S| = \tilde{O}(k/\varepsilon)$. It follows from Equation (2) that we have $(1 + \varepsilon)$-APCs of size $\tilde{O}(k/\varepsilon^3)$.

### 1.4. Organization of the Paper

Section 2 presents sensitivity sampling and our concentration bound (Theorem 2.2) for this algorithm. Sensitivity sampling is the core subroutine used in all of our APC constructions. Section 3 gives a $(1 + \varepsilon)$-APC for Euclidean $k$-means of size $\tilde{O}(k\varepsilon^{-3})$, and Section 4 gives a $(4 + \varepsilon)$-APC for arbitrary metrics of size $\tilde{O}(k\varepsilon^{-2})$ together with an FPT approximation algorithm. We complement these results with a lower bound for $\beta < 4$ and conclude with experiments in Section 5.

## 2. Sensitivity Sampling

We will use the Sensitivity Sampling Algorithm given below as the sampling primitive in all of our APC constructions. The algorithm first computes a constant-factor $k$-means solution. It then defines a sampling distribution $\mu$ based on this clustering, draws $m$ i.i.d. samples, and assigns inverse-probability weights.

---

**Algorithm 1** $\text{SENSAMPLE}(\mathcal{M}, P, k, m)$

---

**Input:** A metric space $\mathcal{M} = (X, d)$, a set of $n$ points $P \subseteq X$, integers $k$ and $m$.

1: Compute a $O(1)$-approximate $k$-means solution $A = \{a_1, \ldots, a_k\}$ for $P$ with centers in $X$. Let $C_j \subset P$ be the cluster centered at $a_j$. For a point $p$ in $C_j$, let $\Delta_p := \text{cost}(C_j, A)/|C_j|$ denote the average cost of $C_j$.

2: Let $\mu : P \to \mathbb{R}^+$ be the following probability distribution. For a point $p \in C_j$,

$$\mu(p) := \frac{1}{4} \cdot \left( \frac{1}{k|C_j|} + \frac{\text{cost}(p, A)}{k\,\text{cost}(C_j, A)} + \frac{\text{cost}(p, A)}{\text{cost}(P, A)} + \frac{\Delta_p}{\text{cost}(P, A)} \right).$$

3: For $i$ from 1 to $m$:

   • Sample point $q_i$ independently from $\mu$.

   • Add $q_i$ to $\Omega$ with weight $w(q_i) := 1/(m \cdot \mu(q_i))$.

**Output:** The set of points $\Omega = \{q_1, \ldots, q_m\}$ and the weights $\{w(q_1), \ldots, w(q_m)\}$.

---

**Remark.** We would also like to apply sensitivity sampling to weighted inputs. We interpret this by expanding each point $p$ of weight $w(p)$ into $w(p)$ unit-weight copies; equivalently, the sampling probability of $p$ is scaled by $w(p)$, and a sampled point receives weight $w(p)/(m\mu(p))$.

It is known from previous work that sensitivity sampling can be used to construct strong coresets.

**Theorem 2.1** (Theorem 1 from (Bansal et al., 2024)). *Sensitivity sampling yields a strong coreset of size $\tilde{O}(k/\varepsilon^4)$ points in Euclidean space for the $k$-means problem with constant probability.*

In this paper we prove a new concentration inequality for the same sampling procedure, which provides uniform cost preservation over an arbitrary finite family of solutions. This theorem is a key black box in our APC analysis; its proof is deferred to Section A.

**Theorem 2.2.** *Let $\Omega$ be a coreset of size $m$ obtained by sensitivity sampling. Suppose $\mathcal{S}$ is a finite family of $k$-center sets with $|\mathcal{S}| = N$. For any $\varepsilon, \delta \in (0, 1)$, if*

$$m = C\varepsilon^{-2} \cdot \left( k\log(k/\delta) + \log N \right)$$

*for an absolute constant $C > 0$, then the coreset preserves*

*the cost of all solutions in $\mathcal{S}$ up to a multiplicative $(1 \pm \varepsilon)$ factor with probability at least $(1 - \delta)$.*

In particular, for finite $n$ point metrics, $N = \binom{n}{k} \leq n^k$, yielding the following corollary.

**Corollary 2.3.** *Sensitivity sampling yields a strong coreset for $k$-means in finite metrics with $n$ points of size $O(k\varepsilon^{-2}(\log n + \log \delta^{-1}))$ with probability $1 - \delta$.*

As mentioned earlier, this is the first coreset construction in finite metrics that is optimal with respect to all parameters, see (Cohen-Addad et al., 2022a) for a matching lower bound.

# 3. Approximation Preserving Coresets for Euclidean Space

In this section, we construct $(1 + \varepsilon)$-approximation preserving coresets for Euclidean $k$-means.

**Theorem 3.1.** *For any point set $P \subseteq \mathbb{R}^d$, there is a randomized algorithm that, with constant probability, constructs a $(1 + \varepsilon)$-approximation preserving coreset of size $\tilde{O}(k/\varepsilon^3)$ for Euclidean $k$-means.*

Given any $\alpha$-approximate solution for the coreset, the lifting procedure described below runs in polynomial time (in the coreset size) and returns an $\alpha(1 + \varepsilon)$-approximate solution for $P$ with constant probability.

We prove Theorem 3.1 in two steps. First, we describe the construction of the APC $\Omega$. Second, we give a lifting procedure that maps any approximate solution on $\Omega$ to an $\alpha(1 + \varepsilon)$-approximate solution with respect to $P$.

**APC Construction.** The coreset is constructed in two rounds of sensitivity sampling: first we compute a strong coreset $\Omega'$ for $P$, and then sample the final APC $\Omega$ from $\Omega'$.

---

**Algorithm 2** APPROXIMATION PRESERVING CORESET for Euclidean space

---

**Input:** A set of points $P \subset \mathbb{R}^d$, integer $k$ and $\varepsilon \in (0, 1/2)$. Let $c$ be a sufficiently large constant.
 1: $\Omega' \leftarrow$ SENSAMPLE$(\mathbb{R}^d, P, k, \tilde{O}(k/\varepsilon^4))$
    {Sample a strong coreset for $P$.}
 2: $\Omega \leftarrow$ SENSAMPLE$(\mathbb{R}^d, \Omega', k, c(k/\varepsilon^3) \log(k/\varepsilon))$
    {Sample the APC.}
**Output:** The weighted set $(\Omega, w)$

---

**Witness family and cost preservation.** We define a finite family of structured solutions and show that, with constant probability, $\Omega$ preserves the cost of every solution in this family. This family is rich enough to contain a near-cost-preserving representative for any solution computed on $\Omega$.

Let $\mathcal{M}$ denote the set of all means of multisets of $3/\varepsilon$ points from $\Omega'$:

$$\mathcal{M} := \{\mu(T) : T \text{ is a multiset of } 3/\varepsilon \text{ points from } \Omega'\}.$$

Let $\mathcal{F}$ be the family of all $k$-center solutions whose centers lie in $\mathcal{M}$:

$$\mathcal{F} := \{S \subseteq \mathcal{M} : |S| = k\}.$$

**Lemma 3.2.** *With constant probability, it holds that*

1. *(**Cost Preservation on $\mathcal{M}$-centers**) For all $S \in \mathcal{F}$,*

$$\text{cost}(\Omega, S) \in (1 \pm \varepsilon)\text{cost}(P, S).$$

2. *(**Cost Preservation for an Optimal Solution**) There exists an optimum solution $C^* \in \arg\min_C \text{cost}(P, C)$ such that*

$$\text{cost}(\Omega, C^*) \in (1 \pm \varepsilon)\text{cost}(P, C^*).$$

*Proof.* We first condition on the event that $\Omega'$ is a strong coreset for $P$. By Theorem 2.1, this event holds with constant probability and gives cost preservation from $P$ to $\Omega'$ for all solutions.

We observe that $|\mathcal{M}| \leq |\Omega'|^{3/\varepsilon}$, and hence $|\mathcal{F}| \leq |\Omega'|^{3k/\varepsilon}$. Including a chosen optimum solution $C^*$, the total number of solutions for which the second sampling step should preserve the cost is bounded by $N = |\Omega'|^{3k/\varepsilon} + 1$. Since $|\Omega'| = \text{poly}(k, 1/\varepsilon)$, we have $\log N = O\left(\frac{k}{\varepsilon} \log(k/\varepsilon)\right)$. Thus the sample size used to construct $\Omega$ satisfies the bound of Theorem 2.2. Therefore, with constant probability,

$$\text{cost}(\Omega, S) \in (1 \pm \varepsilon)\text{cost}(\Omega', S)$$

for every $S \in \mathcal{F}$ and for the chosen optimum solution $C^*$.

Combining this with the strong coreset guarantee for $\Omega'$ gives

$$\text{cost}(\Omega, S) \in (1 \pm O(\varepsilon))\text{cost}(P, S)$$

for every required solution $S$. Rescaling $\varepsilon$ by a constant factor completes the proof. $\square$

**Lifting Algorithm.** It remains to show that any solution computed on $\Omega$ can be mapped to a solution in $\mathcal{F}$ without increasing its cost on $\Omega$ by much. We use the following standard subset-mean approximation lemma.

**Lemma 3.3** (Subset mean approximation (Inaba et al., 1994)). *Let $Q \subseteq \mathbb{R}^d$ and let $\mu := \mu(Q)$ be its mean. For any $\varepsilon \in (0, 1)$, if $T$ is a multiset of $3/\varepsilon$ points sampled uniformly at random from $Q$, then*

$$\Pr\left[\text{cost}(Q, \mu(T)) \leq (1 + \varepsilon)\text{cost}(Q, \mu)\right] \geq 2/3.$$

**Algorithm 3** LIFTING for Euclidean Space

**Input:** Weighted coreset $\Omega$ and a set of $k$ centers $S \subseteq \mathbb{R}^d$.
**Output:** Lifted solution $S'$
1: Compute the clusters $(C_1, \ldots, C_k)$ induced by $S$ on $\Omega$.
2: For each cluster $C_i$, compute its weighted mean $\mu_i$.
3: **for** $i = 1$ to $k$ **do**
4:     Repeat $t := \log_3(3k)$ times:
5:        Sample $3/\varepsilon$ points from $C_i$ with replacement according to the normalized weights in $C_i$, and let $\widehat{\mu}$ be their mean.
6:     Choose the sampled mean $\widehat{\mu}$ minimizing $\|\mu_i - \widehat{\mu}\|^2$, and set $s'_i := \widehat{\mu}$.
7: **end for**
8: **return** $S' = (s'_1, \ldots, s'_k)$

Given a solution $S$ for $\Omega$, the lifting algorithm clusters the points of $\Omega$ according to $S$. For each cluster, it samples $3/\varepsilon$ points according to the normalized weights and replaces the corresponding center by their mean. Since $\Omega \subseteq \Omega'$, every new center lies in $\mathcal{M}$.

**Lemma 3.4.** *With probability at least $2/3$, Algorithm 3 maps any solution $S$ to a solution $S' \in \mathcal{F}$ such that*

$$\text{cost}(\Omega, S') \le (1 + \varepsilon)\text{cost}(\Omega, S).$$

*Proof.* Fix a cluster $C_i$ induced by $S$ on $\Omega$, and let $\mu_i$ be its weighted mean. By Lemma 3.3, one sample of $3/\varepsilon$ points from $C_i$ according to the normalized weights gives a mean $\widehat{\mu}$ with $\text{cost}(C_i, \widehat{\mu}) \le (1 + \varepsilon)\text{cost}(C_i, \mu_i)$ with probability at least $2/3$.

The algorithm repeats this experiment $t = \log_3(3k)$ times and keeps the sampled mean closest to $\mu_i$, equivalently the sampled mean with minimum cost on $C_i$. Thus cluster $C_i$ fails with probability at most $(1/3)^t$. A union bound over all clusters gives failure probability at most $k(1/3)^t \le 1/3$. Therefore, with probability at least $2/3$, the total cost increases by at most a $(1 + \varepsilon)$ factor.

Finally, each center of $S'$ is the mean of $3/\varepsilon$ points from $\Omega$. Since $\Omega \subseteq \Omega'$, each center lies in $\mathcal{M}$; hence $S' \in \mathcal{F}$. $\square$

**Proof of Theorem 3.1.** The lifting algorithm runs in $\text{poly}(k, 1/\varepsilon, d)$ time, since it operates only on the coreset $\Omega$ of size $\tilde{O}(k/\varepsilon^3)$ and performs $O(\log k)$ trials of size $O(1/\varepsilon)$ for each of the $k$ clusters.

Let $S$ be an $\alpha$-approximate solution for $\Omega$, and let $S' = \text{LIFTING}(S)$ be the solution returned by Algorithm 3. Let $C^*$ be the optimum solution for $P$ whose cost is preserved by Lemma 3.2. Conditioning on the events of Lemmas 3.2

and 3.4, we have

$$
\begin{aligned}
\text{cost}(P, S') &\le \frac{1}{1 - \varepsilon}\text{cost}(\Omega, S') && \text{(via Lemma 3.2)} \\
&\le \frac{1 + \varepsilon}{1 - \varepsilon}\text{cost}(\Omega, S) && \text{(via Lemma 3.4)} \\
&\le \frac{\alpha(1 + \varepsilon)}{1 - \varepsilon} \min_C \text{cost}(\Omega, C) \\
& && \text{(since } S \text{ is } \alpha\text{-approximate for } \Omega) \\
&\le \frac{\alpha(1 + \varepsilon)}{1 - \varepsilon}\text{cost}(\Omega, C^*) \\
&\le \frac{\alpha(1 + \varepsilon)^2}{1 - \varepsilon}\text{cost}(P, C^*). && \text{(via Lemma 3.2)}
\end{aligned}
$$

Thus $S'$ is an $\alpha(1 + O(\varepsilon))$-approximate solution for $P$. Running the construction with a sufficiently small constant multiple of $\varepsilon$ gives the claimed $\alpha(1 + \varepsilon)$ guarantee. $\square$

# 4. Approximation Preserving Coresets for Arbitrary Metrics

In this section, we construct $(4 + \varepsilon)$-approximation preserving coresets for $k$-means in arbitrary finite metrics.

**Theorem 4.1.** *For any finite metric space $\mathcal{M} = (X, d)$ and any point set $P \subseteq X$, there exists a $(4 + \varepsilon)$-approximation preserving coreset of size $\tilde{O}(k/\varepsilon^2)$.*

We prove the theorem in two steps. First, we describe the APC construction which involves iteratively using sensitivity sampling to compress the input. Then we describe a simple lifting procedure that maps any solution computed on the coreset to a solution for the original instance.

**APC Construction.** Strong coresets for finite metrics have size $O(k\varepsilon^{-2} \log n)$, where $n = |P|$. To remove the dependence on $n$, we iteratively downsample. Starting from $\Omega_0 = P$, the coreset $\Omega_i$ is sampled from $\Omega_{i-1}$ and is required to preserve the costs of all solutions whose centers lie in the support of $\Omega_{i-1}$. As the support size decreases, the number of such candidate solutions also decreases, so the next coreset can be smaller. Repeating this procedure yields a coreset of size $\tilde{O}(k\varepsilon^{-2})$. This is similar to the iterative size reduction idea of (Braverman et al., 2021).

For an integer $i \ge 0$, define the $i$-fold iterated logarithm by $\log^{(0)}(n) = n$ and $\log^{(i)}(n) = \log(\log^{(i-1)}(n))$ for $i > 0$.

**Algorithm 4** APPROXIMATION PRESERVING CORESET for Finite Metric Spaces

**Input:** A finite metric space $(X, d)$, point set $P \subseteq X$, integer $k$, accuracy $\varepsilon \in (0, 1)$.
1: $\Gamma \leftarrow C \cdot k\varepsilon^{-2} \log(k/\varepsilon)$, where $C > 0$ is a sufficiently large constant.
2: Initialize $(\Omega_0, w_0) \leftarrow (P, \mathbf{1})$.
3: Let $B > 1$ be a sufficiently large absolute constant.
4: $t \leftarrow \min\{i \geq 1 : \log^{(i)}(n) \leq B\}$, where $n := |P|$.
5: **for** $i = 1$ **to** $t$ **do**
6: $\quad L_i \leftarrow \max\{\log^{(i)}(n), B\}$.
7: $\quad m_i \leftarrow \Gamma L_i^2$.
8: $\quad (\Omega_i, w_i) \leftarrow \textsc{SenSample}\big((X, d), (\Omega_{i-1}, w_{i-1}), k, m_i\big)$.
9: **end for**
10: $(\Omega, w) \leftarrow (\Omega_t, w_t)$.
**Output:** The weighted set $(\Omega, w)$.

The main property of the construction is the following.

**Lemma 4.2.** *The coreset $\Omega$ output by Algorithm 4 has size $\tilde{O}(k\varepsilon^{-2})$. Moreover, with constant probability, it satisfies:*

1. *(**Cost Preservation on $\Omega$-centers**) For every $S \subseteq \Omega$ with $|S| = k$,*

$$\mathrm{cost}(\Omega, S) \in (1 \pm \varepsilon)\mathrm{cost}(P, S).$$

2. *(**Cost Preservation for an Optimal Solution**) There exists an optimal solution $C^* \subseteq X$ for $P$ such that*

$$\mathrm{cost}(\Omega, C^*) \in (1 \pm \varepsilon)\mathrm{cost}(P, C^*).$$

*Proof.* Let $S^* \subseteq X$ be a fixed optimal solution for $P$. For each $i \in [t]$, define

$$L_i := \max\{\log^{(i)}(n), B\}, \quad \varepsilon_i := \varepsilon L_i^{-1/2}, \quad \delta_i := \frac{1}{4m_{i-1}},$$

where $m_0 := n$.

Fix the randomness of $\Omega_1, \ldots, \Omega_{i-1}$. At step $i$, we need to preserve the costs of all $k$-center sets contained in $\Omega_{i-1}$, together with $S^*$. Thus the number of relevant solutions is at most

$$N_i \leq |\Omega_{i-1}|^k + 1 \leq 2m_{i-1}^k.$$

By Theorem 2.2, it suffices to sample

$$\Omega\big(\varepsilon_i^{-2}\left(k \log k + \log N_i + k \log(1/\delta_i)\right)\big)$$

points. Since $\log m_{i-1} = O(\log(k/\varepsilon) + L_i)$, we have

$$\varepsilon_i^{-2}\left(k \log k + \log N_i + k \log(1/\delta_i)\right)$$
$$= O\big(k\varepsilon^{-2}L_i(\log(k/\varepsilon) + L_i)\big) = O(\Gamma L_i^2) = O(m_i),$$

for a sufficiently large choice of the constant $C$ in $\Gamma$. Therefore, with probability at least $1 - \delta_i$, $\Omega_i$ preserves all relevant costs up to a factor $(1 \pm \varepsilon_i)$.

By a union bound over the iterations, all stepwise preservation events hold simultaneously with constant probability. Condition on this event. If $S \subseteq \Omega = \Omega_t$ with $|S| = k$, then $S \subseteq \Omega_{i-1}$ for every $i$, so the stepwise guarantees apply to $S$ throughout the sequence. Hence

$$\mathrm{cost}(\Omega, S) \in \left(\prod_{i=1}^{t}(1 \pm \varepsilon_i)\right)\mathrm{cost}(P, S).$$

The same argument applies to $S^*$. Since the iterated logarithms decrease rapidly and $B$ is a sufficiently large constant, $\sum_{i=1}^{t} \varepsilon_i = O(\varepsilon)$, so the product error is $1 \pm O(\varepsilon)$.

Finally, by the stopping rule, $L_t = B$, and therefore

$$|\Omega| = m_t = \Gamma B^2 = \tilde{O}(k\varepsilon^{-2}).$$

Rescaling $\varepsilon$ by constant factors completes the proof. $\square$

**Lifting Algorithm.** It remains to explain how to convert an $\alpha$-approximate solution for $\Omega$ into a solution which is $(4 + \varepsilon)\alpha$-approximate solution for $P$. The lifting procedure is simple: each center is moved to its nearest point in $\Omega$. The following standard lemma bounds the cost increase.

**Lemma 4.3.** *Let $(X, d)$ be a metric space and let $Q \subseteq X$. For any set of $k$ centers $S = \{s_1, \ldots, s_k\} \subseteq X$, let $S' = \{s'_1, \ldots, s'_k\} \subseteq Q$, where each $s'_i$ is the point of $Q$ closest to $s_i$. Then*

$$\mathrm{cost}(Q, S') \leq 4\mathrm{cost}(Q, S).$$

*Proof.* For any $q \in Q$, let $s_i$ be the nearest center to $q$ in $S$. Since $s'_i$ is the closest point of $Q$ to $s_i$, we have

$$d(q, s'_i) \leq d(q, s_i) + d(s_i, s'_i) \leq 2d(q, s_i).$$

Squaring and summing over $q \in Q$ gives the claim. $\square$

**Algorithm 5** LIFTING for Finite Metrics

**Input:** Weighted coreset $(\Omega, w)$ and centers $\widehat{S} = \{s_1, \ldots, s_k\} \subseteq X$.
1: For each $i$, let $s'_i$ be the point in $\Omega$ closest to $s_i$.
**Output:** $\widehat{S}' = \{s'_1, \ldots, s'_k\} \subseteq \Omega$.

The proof of Theorem 4.1 follows the same inequality chain as Theorem 3.1. The only difference is the lifting step: by Lemma 4.3, mapping each center to its nearest point in $\Omega$ increases the coreset cost by at most a factor of 4. Lemma 4.2 then transfers the lifted solution and the optimum solution between $\Omega$ and $P$, giving a $(4 + O(\varepsilon))$-APC. Rescaling $\varepsilon$ completes the proof.

## 4.1. FPT Algorithm

We record an algorithmic consequence of Theorem 4.1.

**Theorem 4.4.** *For finite metric $k$-means, there is a randomized $(4 + \varepsilon)$-approximation algorithm with running time near-linear in the input size plus $(\varepsilon^{-1} \log k)^{O(k)}$.*

*Proof.* Construct the coreset $\Omega$ from Algorithm 4, and let $m := |\Omega| = \tilde{O}(k\varepsilon^{-2})$. Enumerate all $k$-subsets of $\Omega$, and let $\widehat{S}$ be the minimum-cost such solution on $\Omega$.

Let $C^*$ be an optimal solution for $P$ whose cost is preserved by Lemma 4.2. By Lemma 4.3, there exists a set $S_\Omega \subseteq \Omega$ of $k$ centers such that

$$\text{cost}(\Omega, S_\Omega) \leq 4 \min_{C \subseteq X, \, |C| = k} \text{cost}(\Omega, C).$$

Since $\widehat{S}$ is the best $k$-subset of $\Omega$, we have

$$\text{cost}(\Omega, \widehat{S}) \leq \text{cost}(\Omega, S_\Omega) \leq 4\text{cost}(\Omega, C^*).$$

Using Lemma 4.2 for both $\widehat{S} \subseteq \Omega$ and $C^*$,

$$\text{cost}(P, \widehat{S}) \leq \frac{1}{1 - \varepsilon}\text{cost}(\Omega, \widehat{S}) \leq \frac{4}{1 - \varepsilon}\text{cost}(\Omega, C^*)$$
$$\leq \frac{4(1 + \varepsilon)}{1 - \varepsilon}\text{cost}(P, C^*).$$

Thus $\widehat{S}$ is a $(4 + O(\varepsilon))$-approximation for $P$.

The enumeration takes

$$O\left(\binom{m}{k}\text{poly}(m)\right) = (\varepsilon^{-1} \log k)^{O(k)}.$$

Adding the coreset construction time gives the claimed running time. $\qquad\square$

## 4.2. Lower bound

We prove a lower bound for APCs where the summary consists only of the retained input points. In this model, if two inputs induce the same retained subset, then no post-processing algorithm using only the coreset and the underlying metric space can distinguish them.

First we show a combinatorial packing lemma used in the lower bound.

**Lemma 4.5.** *For every fixed $0 < \tau < 1$, there are constants $\gamma > 0$ and $n_0$ such that, for all $n \geq n_0$ and all $r \leq \tau n/4$, there exists a family $\mathcal{F} \subseteq \binom{[n]}{r}$ satisfying $|\mathcal{F}| \geq \exp\left(\gamma r \log \frac{n}{r}\right)$, and for every pair of distinct sets $S, T \in \mathcal{F}$, $|S \cap T| \leq \tau r$.*

*Proof.* The proof is by a standard application of the probabilistic method. Let $m = \exp\left(\gamma r \log(n/r)\right)$, where $\gamma > 0$

is a sufficiently small constant depending only on $\tau$, and choose $m$ sets independently and uniformly at random from $\binom{[n]}{r}$. For a fixed set $S$, the random variable $|S \cap T|$, where $T$ is uniformly distributed over $\binom{[n]}{r}$, has a hypergeometric distribution with mean $r^2/n$. Since $r \leq \tau n/4$, a standard hypergeometric tail bound (Hoeffding, 1963) gives

$$\Pr[|S \cap T| > \tau r] \leq \exp\left(-cr \log \frac{n}{r}\right)$$

for some constant $c > 0$.

Therefore, by a union bound over all pairs of sampled sets, the probability that all pairs $S, T$ satisfy $|S \cap T| \leq \tau r$ is at least $1 - \binom{m}{2}\exp(-cr \log(n/r))$ which is positive for small enough $\gamma$. $\qquad\square$

**Theorem 4.6.** *There exists a metric space $(X, d)$ with $N$ points and an input set $P \subseteq X$, such that any $\beta$-approximation preserving coreset for $P$ with $\beta < 4$ has size $\Omega(\log N / \log \log N)$.*

*Proof.* We prove the result for $k = 1$. We construct the metric space $\mathcal{M} = (X, d)$ containing a set of possible input points $A = \{p_1, p_2, \ldots, p_n\}$.

Fix a sufficiently small constant $\tau > 0$. We use Lemma 4.5 with $r = \sqrt{n}$ and get a set family $\mathcal{F}$ of size $|\mathcal{F}| \geq \exp(\Omega(r \log(n/r))) = \exp(\Omega(\sqrt{n} \log n))$ such that for any $S, T \in \mathcal{F}$, we have $|S \cap T| \leq \tau r$.

The metric instance is as follows. For every set $S \in \mathcal{F}$, we add a candidate center $c_S$. Thus, we have $X = A \cup \{c_S : S \in \mathcal{F}\}$. We define the relevant distances by

$$d(p_i, c_S) = \begin{cases} 1 & \text{if } i \in S, \\ 2 & \text{if } i \notin S. \end{cases}$$

Moreover, we set $d(p_i, p_j) = 2$ for all $i \neq j$, and $d(c_S, c_T) = 2$ for all distinct $S, T \in \mathcal{F}$. These distances define a metric as all distances lie in $\{1, 2\}$.

Assuming all coreset sizes are of size $s \leq r$, using a pigeonhole argument, we observe that there exist two sets $S, T \in \mathcal{F}$ that have the same coreset $Q$. This is because $|\mathcal{F}| = \exp(\Omega(r \log(n/r))) > \sum_{j \leq s} \binom{n}{j}$.

However, no single center can be a $\beta$-approximation for both $S$ and $T$ when $\beta < 4$, provided $\tau$ is sufficiently small. Indeed, $c_S$ has cost at least $(4 - 3\tau)r$ on $T$, $c_T$ has cost at least $(4 - 3\tau)r$ on $S$, and any other candidate center is bad for both by the same packing argument. Moreover, any data point $p_i \in A$ has cost at least $4(r - 1)$ on either instance. Thus the common output fails to achieve factor $\beta$ on at least one of $S$ and $T$ and we get that the coreset size of one of them must be at least $\Omega(r)$.

Finally, $N = n + |\mathcal{F}| = \exp(\Theta(\sqrt{n} \log n))$. Thus $r = \sqrt{n} = \Theta\left(\frac{\log N}{\log \log N}\right)$, which gives us the lower bound. $\qquad\square$

# 5. Experimental Evaluation

In this section, we give a detailed overview of our experimental results. Codebase and results can be found here. Google Colab CPU was used to conduct the experiments. The goal is to analyze the performance of some prominent approximation algorithms for the $k$-means problem and to which degree the approximation preserving guarantee holds for real world datasets.

**Datasets.** We use three popular datasets - Adult, MNIST and CoverType for the experiments.

**Algorithms.** We test the performance of APCs using three approximation algorithms K-MEANS++, LOCAL-SEARCH and LOCAL-SEARCH++. K-MEANS++ is an initialization method that samples centers sequentially from a distribution, favoring points far from the current centers, after which Lloyd's iterations are run to improve the solution. LOCAL-SEARCH starts from an initial solution and repeatedly attempts improving single-center swaps (replacing one center by a data point). LOCAL-SEARCH++ (Lattanzi & Sohler, 2019; Choo et al., 2020) follows the same swap-based improvement scheme, but proposes swap candidates via the same $D^2$ (cost-biased) sampling used in k-means++, focusing the search on points with high cost.

As a baseline, we execute the algorithms on the full dataset and obtain approximate solutions. To analyze the performance of APCs, we construct these coresets and for each dataset compare the approximate solutions obtained on the coresets with the ones obtained earlier.

**Setup.** Let $m$ denote the number of points in our coreset. We run the experiments with parameters $k = \{2, 5, 10, 50, 100\}$ and $m = \{2k, 5k, 10k, 50k, 100k\}$. For each choice of parameters, we repeat each experiment 10 times and compute the averages of the output parameters.

**Results.** We plot the values of *average cost ratio* in terms of $m/k$ (see Figure 1). As $k$ is a lower bound on the number of coreset points, we examine how the approximation factor improves after only adding $O(k)$ points to the coreset.

Across all datasets and algorithms, the cost–ratio curves as a function of $m/k$ are visually consistent across different choices of $k$. Therefore, for each value of $m/k$ we aggregate the results by averaging the ratio over all $k$ values and trial runs, and plot these averages. See Appendix B for further details.

As expected, as $m/k$ increases, the accuracy of the solution obtained on the coreset increases, with the ratio approaching 1. Sampling $10k$ points already leads to only a 1.2 factor loss in approximation as compared to the full dataset.

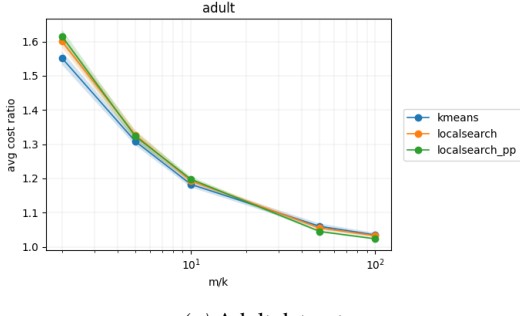

*(a)* Adult dataset

*Figure 1.* Average cost ratio vs. $m/k$ (Adult dataset) for each algorithm: for each run we take the clustering cost on the full dataset using centers learned from the coreset, divide by the cost on the full dataset using centers learned from the full dataset, and then average this ratio over all $k$ and trials.

In addition, we plot the *ratio of the running times* for running the approximation algorithms on the coresets and the full dataset.

Due to space constraints, we just give the plot for one of the datasets. We observe substantial speedups from using coresets, especially for LOCAL-SEARCH and LOCAL-SEARCH++: runtimes drop by roughly $20\times$–$80\times$. As expected, the speedup diminishes as the coreset size increases.

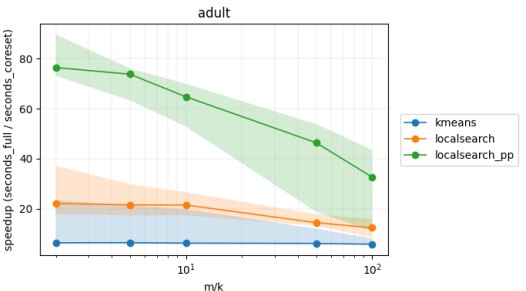

*Figure 2.* Speedup from using coresets vs. $m/k$: for each dataset and algorithm, we report the ratio between the runtime of running the algorithm directly on the full dataset and the coreset

Finally, we report a table of costs for the three algorithms K-MEANS++, LOCALSEARCH, and LOCALSEARCH++ across all choices of $k$, $m$, and datasets. The point is to demonstrate that our APC guarantees are largely insensitive to the quality of the underlying approximation routine: even when an algorithm produces better or worse solutions on the full data, the corresponding solution obtained via the coreset remains comparably good, indicating that the APC is effectively *universal* across algorithms. The details are relegated to the appendix.

## Acknowledgements

C.S. and S.S. are supported by a Google Research Award.

## Impact Statement

This paper presents work whose goal is to advance the field of Machine Learning. There are many potential societal consequences of our work, none which we feel must be specifically highlighted here.

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

## A. Sensitivity Sampling

We briefly recall the notation used throughout this section. Let $A = \{a_1, \ldots, a_k\}$ denote an $O(1)$-approximate $k$-means solution for $P$, and let $C_1, \ldots, C_k$ be the induced clustering, where $C_j = \{p \in P : a_j$ is the closest center to $p\}$. For each cluster $C_j$, we write $\text{cost}(C_j, A) = \sum_{p \in C_j} \text{cost}(p, A)$ and $\Delta_j := \text{cost}(C_j, A)/|C_j|$ for its average cost.

The coreset $\Omega = \{q_1, \ldots, q_m\}$ with weights $\{w(q_1), \ldots, w(q_m)\}$ is obtained by the sensitivity sampling algorithm defined previously (see Algorithm 1). For any subset $T \subseteq P$ and any set of centers $S$, we define

$$\text{cost}_\Omega(T, S) := \sum_{q \in \Omega \cap T} w(q) \, \text{cost}(q, S).$$

Our main goal is to prove the following theorem.

**Theorem 2.2.** *Let $\Omega$ be a coreset of size $m$ obtained by sensitivity sampling. Suppose $\mathcal{S}$ is a finite family of $k$-center sets with $|\mathcal{S}| = N$. For any $\varepsilon, \delta \in (0, 1)$, if*
$$m = C\varepsilon^{-2} \cdot \left(k \log(k/\delta) + \log N\right)$$
*for an absolute constant $C > 0$, then the coreset preserves the cost of all solutions in $\mathcal{S}$ up to a multiplicative $(1 \pm \varepsilon)$ factor with probability at least $(1 - \delta)$.*

**Setup and parameter choice.** For the remainder of this section, we fix the coreset size to be
$$m = C\varepsilon^{-2} \cdot \left(k \log(k/\delta) + \log N\right),$$
for a sufficiently large absolute constant $C$, and analyze the guarantees of the resulting coreset.

**Proof roadmap.** We prove Theorem 2.2 in two steps. First, we establish a key structural property of the sampled coreset—valid for general metric spaces: with high probability it preserves, for every approximate-$k$-means cluster $C_j$, both (i) the total coreset weight assigned to $C_j$ and (ii) the weighted cost of points in $C_j$ to the approximate solution $A$ (Event $E$). This per-cluster preservation will be used as a black box throughout the proof. Second, fixing any candidate solution $S \in \mathcal{S}$, we partition clusters into *close* and *far* depending on the distance from their representative center $a_j$ to $S$. For close clusters we bound the approximation error by concentration (Bernstein) and take a union bound over $S \in \mathcal{S}$, while far clusters are controlled deterministically using Event $E$. Combining the two bounds and union bounding completes the proof.

**Structural properties of the coreset.** We first record a cluster-level event satisfied by sensitivity sampling.

**Definition A.1** (Event $E$). The coreset $\Omega$ satisfies event $E$ if for every cluster $C_j$,

$$\sum_{q \in C_j \cap \Omega} w(q) = (1 \pm \varepsilon) \, |C_j|, \qquad \sum_{q \in C_j \cap \Omega} w(q)\text{cost}(q, A) = (1 \pm \varepsilon) \, \text{cost}(C_j, A).$$

**Claim A.2.** *The coreset $\Omega$ satisfies event $E$ with probability at least $1 - \delta/4$.*

This property follows from the design of the sensitivity sampling distribution, which guarantees sufficient sampling from every cluster while controlling the variance contributed by points of different costs. The formal proof is a standard application of Bernstein's inequality and presented later in Section A.4.

In addition to the coreset properties above, we will repeatedly use the following simple geometric bound. It relates the cost of an arbitrary solution $S$ evaluated at cluster representatives of the approximate solution $A$ to the true cost $\text{cost}(P, S)$.

**Claim A.3.** *For any set of centers $S$,*

$$\sum_{j \in [k]} |C_j|\text{cost}(a_j, S) \lesssim \text{cost}(P, S).$$

*Proof.* For any point $p \in C_j$, the approximate triangle inequality gives

$$\text{cost}(a_j, S) \leq 2(\text{cost}(a_j, p) + \text{cost}(p, S)).$$

Summing this inequality over all $p \in C_j$ and then over all clusters gives

$$\sum_{j=1}^{k} |C_j| \text{cost}(a_j, S) \lesssim \text{cost}(P, A) + \text{cost}(P, S).$$

Since $A$ is an $O(1)$-approximate solution, $\text{cost}(P, A) \lesssim \text{cost}(P, S)$, completing the proof. $\qquad\square$

### A.1. Proof of Theorem 2.2

Proving Theorem 2.2 reduces to showing that

$$\Pr_{\Omega}\left[\sup_{S \in \mathcal{S}} \left| \frac{\text{cost}(P, S) - \text{cost}_{\Omega}(P, S)}{\text{cost}(P, S)} \right| \geq \varepsilon \right] \leq \delta. \tag{3}$$

**Far and close clusters.** Fix $S \in \mathcal{S}$. We partition clusters based on the distance of cluster representatives from $S$.

**Definition A.4** (Far and close clusters). A cluster $C_j$ is *far* from $S$ if $\text{cost}(a_j, S) > \Delta_j \varepsilon^{-2}$, and *close* otherwise.

Accordingly, we write $P_F(S)$ for the points belonging to clusters that are far from $S$, and $P_C(S)$ for the points belonging to close clusters.

Far clusters are easy to handle using the cluster-level structural guarantees provided by sensitivity sampling; their contribution to the total cost can be controlled directly. The proof of the following lemma is presented later.

**Lemma A.5** (Far clusters). *With probability at least $1 - \delta/2$ over the randomness of the coreset,*

$$\sup_{S \in \mathcal{S}} \left| \frac{\text{cost}(P_F(S), S) - \text{cost}_{\Omega}(P_F(S), S)}{\text{cost}(P, S)} \right| \leq \varepsilon/2. \tag{4}$$

The remaining work is therefore to bound the error contributed by close clusters. We do this next.

**Lemma A.6** (Close clusters). *With probability at least $1 - \delta/2$ over the randomness of the coreset,*

$$\sup_{S \in \mathcal{S}} \left| \frac{\text{cost}(P_C(S), S) - \text{cost}_{\Omega}(P_C(S), S)}{\text{cost}(P, S)} \right| \leq \varepsilon/2.$$

By a union bound, Lemmas A.5 and A.6 hold simultaneously with probability at least $1 - \delta$, and together establish Theorem 2.2.

### A.2. Handling Close Clusters

We now prove Lemma A.6. Fix a solution $S \in \mathcal{S}$. Our goal is to control the contribution of points belonging to clusters that are close to $S$.

Recall that $P_C(S)$ denotes the set of points in clusters that are close to $S$. We decompose their total cost to $S$ into two components: a term that depends only on the distances between cluster representatives and $S$, and a residual term that captures the variation of individual points within each cluster. Formally, we decompose

$$\text{cost}(P_C(S), S) = A(S) + B(S),$$

where

$$A(S) := \sum_{p \in P_C(S)} \big( \text{cost}(p, S) - \text{cost}(a(p), S) \big),$$

$$B(S) := \sum_{p \in P_C(S)} \text{cost}(a(p), S).$$

We analyze these two terms separately. Let $\Omega$ denote the random coreset produced by sensitivity sampling. The contribution of close-cluster points in the coreset admits an analogous decomposition:

$$\text{cost}_\Omega(P_C(S), S) = C(S, \Omega) + D(S, \Omega),$$

where

$$C(S, \Omega) := \sum_{q_i \in \Omega \cap P_C(S)} w(q_i) \cdot \big(\text{cost}(q_i, S) - \text{cost}(a(q_i), S)\big)$$

$$D(S, \Omega) := \sum_{q_i \in \Omega \cap P_C(S)} w(q_i) \cdot \text{cost}(a(q_i), S).$$

By construction of the coreset weights, $C(S, \Omega)$ and $D(S, \Omega)$ are unbiased estimators of $A(S)$ and $B(S)$, respectively. We show that, for the choice of $m$ specified in Lemma A.6, both estimators concentrate sufficiently well.

**Claim A.7.** *With probability at least $1 - \delta/4$,*

$$\sup_{S \in \mathcal{S}} \left| \frac{A(S) - C(S, \Omega)}{\text{cost}(P, S)} \right| \le \varepsilon/4.$$

**Claim A.8.** *With probability at least $1 - \delta/4$,*

$$\sup_{S \in \mathcal{S}} \left| \frac{B(S) - D(S, \Omega)}{\text{cost}(P, S)} \right| \le \varepsilon/4.$$

The benefit of this decomposition is twofold: the center term $D(S, \Omega)$ is controlled directly using the cluster-level structural guarantees of sensitivity sampling, while the remainder term $C(S, \Omega)$ admits strong variance bounds, allowing us to apply concentration inequalities. We begin by proving Claim A.8.

*Proof of Claim A.8.* Let $\mathcal{J}_C(S)$ be the set of clusters that are close to $S$. Then

$$B(S) = \sum_{j \in \mathcal{J}_C(S)} |C_j| \text{cost}(a_j, S).$$

Condition on Event $E$. For every cluster $C_j$, the total weight of coreset points in $C_j$ is $(1 \pm \varepsilon)|C_j|$. Hence

$$D(S, \Omega) = \sum_{j \in \mathcal{J}_C(S)} (1 \pm \varepsilon)|C_j| \text{cost}(a_j, S).$$

It follows that

$$|B(S) - D(S, \Omega)| \le \varepsilon \sum_{j \in \mathcal{J}_C(S)} |C_j| \text{cost}(a_j, S) \le \varepsilon \sum_{j=1}^{k} |C_j| \text{cost}(a_j, S) \lesssim \varepsilon \, \text{cost}(P, S),$$

where the last inequality follows from Claim A.3. Since Event $E$ holds with probability at least $1 - \delta/4$, the claim follows after rescaling constants. $\square$

### A.3. Proof of Claim A.7

We prove Claim A.7. Fix $S \in \mathcal{S}$. The goal is to control the deviation of the coreset estimator $C(S, \Omega)$ from its expectation $A(S)$. We do this by writing the normalized error as a sum of independent random variables and applying Bernstein's inequality.

For each sampled point $q_i$, define

$$X_i(S, \Omega) := \frac{w(q_i)\big(\text{cost}(q_i, S) - \text{cost}(a(q_i), S)\big)\mathbf{1}[q_i \in P_C(S)]}{\text{cost}(P, S)}.$$

Then

$$\sum_{i=1}^{m} X_i(S, \Omega) = \frac{C(S, \Omega)}{\text{cost}(P, S)}, \qquad \mathbb{E}\left[\sum_{i=1}^{m} X_i(S, \Omega)\right] = \frac{A(S)}{\text{cost}(P, S)}.$$

Thus, proving Claim A.7 amounts to bounding the deviation of $\sum_i X_i(S, \Omega)$ from its expectation, uniformly over $S \in \mathcal{S}$.

**Bernstein's inequality.** If $X_1, \ldots, X_m$ are independent random variables with $|X_i - \mathbb{E}\, X_i| \leq M$, then for every $t \geq 0$,

$$\Pr\left[\left|\sum_{i=1}^{m}(X_i - \mathbb{E}\, X_i)\right| > t\right] \leq 2\exp\left(-\frac{t^2}{2\sum_{i=1}^{m}\mathrm{Var}(X_i) + \frac{2}{3}Mt}\right).$$

Thus, it remains to bound the total variance and the maximum deviation of one summand. We will use the following geometric inequalities.

**Claim A.9.** *For any point $p \in P$, the following hold.*

1. $\left|\mathrm{cost}(p, S) - \mathrm{cost}(a(p), S)\right| \lesssim \sqrt{\mathrm{cost}(p, S)\mathrm{cost}(p, A)} + \mathrm{cost}(p, A)$.

2. *If $p$ lies in a cluster that is close to $S$, then $\left|\mathrm{cost}(p, S) - \mathrm{cost}(a(p), S)\right| \lesssim \varepsilon^{-1}\left(\Delta_p + \mathrm{cost}(p, A)\right)$.*

*Proof.* Let $a = a(p)$. By the triangle inequality,

$$|\mathrm{dist}(p, S) - \mathrm{dist}(a, S)| \leq \mathrm{dist}(p, a), \qquad \mathrm{dist}(a, S) \leq \mathrm{dist}(p, S) + \mathrm{dist}(p, a).$$

Using $|u^2 - v^2| = (u + v)|u - v|$, we get

$$\begin{aligned}
\left|\mathrm{cost}(p, S) - \mathrm{cost}(a, S)\right| &= |\mathrm{dist}^2(p, S) - \mathrm{dist}^2(a, S)| \\
&\leq (2\mathrm{dist}(p, S) + \mathrm{dist}(p, a))\mathrm{dist}(p, a) \\
&\lesssim \sqrt{\mathrm{cost}(p, S)\mathrm{cost}(p, A)} + \mathrm{cost}(p, A),
\end{aligned}$$

which proves the first part.

For the second part, suppose $p \in C_j$ and $C_j$ is close to $S$. Then $\mathrm{cost}(a_j, S) \leq \Delta_j \varepsilon^{-2}$, and hence

$$\mathrm{cost}(p, S) \leq 2\mathrm{cost}(p, A) + 2\mathrm{cost}(a_j, S) \leq 2\mathrm{cost}(p, A) + 2\Delta_j \varepsilon^{-2}.$$

Substituting this into the first part gives

$$\left|\mathrm{cost}(p, S) - \mathrm{cost}(a, S)\right| \lesssim \varepsilon^{-1}\sqrt{\Delta_j \mathrm{cost}(p, A)} + \mathrm{cost}(p, A) \lesssim \varepsilon^{-1}\left(\Delta_j + \mathrm{cost}(p, A)\right),$$

where we used $\sqrt{uv} \leq (u + v)/2$ and $\Delta_p = \Delta_j$. $\qquad\square$

**Claim A.10.** *For fixed $S \in \mathcal{S}$, the random variables $X_i = X_i(S, \Omega)$ satisfy*

$$\sum_{i=1}^{m}\mathrm{Var}(X_i) \lesssim \frac{1}{m}, \qquad |X_i - \mathbb{E}\, X_i| \lesssim \frac{1}{\varepsilon m}.$$

*Proof.* By definition,

$$\mathrm{Var}(X_i) \leq \mathbb{E}[X_i^2] = \frac{1}{m^2\mathrm{cost}(P, S)^2}\sum_{p \in P_C(S)}\frac{\left(\mathrm{cost}(p, S) - \mathrm{cost}(a(p), S)\right)^2}{\mu(p)}.$$

Using the first part of Claim A.9 and the sensitivity lower bound $\mu(p) \gtrsim \mathrm{cost}(p, A)/\mathrm{cost}(P, A)$, we obtain

$$\begin{aligned}
\mathrm{Var}(X_i) &\lesssim \frac{1}{m^2\mathrm{cost}(P, S)^2}\sum_{p \in P_C(S)}\frac{\mathrm{cost}(p, S)\mathrm{cost}(p, A) + \mathrm{cost}(p, A)^2}{\mu(p)} \\
&\lesssim \frac{\mathrm{cost}(P, A)}{m^2\mathrm{cost}(P, S)^2}\left(\mathrm{cost}(P, S) + \mathrm{cost}(P, A)\right) \lesssim \frac{1}{m^2},
\end{aligned}$$

where the last step uses that $A$ is an $O(1)$-approximate solution, so $\mathrm{cost}(P, A) \lesssim \mathrm{cost}(P, S)$. Summing over $i$ gives $\sum_i \mathrm{Var}(X_i) \lesssim 1/m$.

For the range bound, condition on $q_i = p \in P_C(S)$. By the second part of Claim A.9,

$$|X_i| \leq \frac{1}{m\mu(p)} \cdot \frac{\varepsilon^{-1}(\Delta_p + \text{cost}(p, A))}{\text{cost}(P, S)}.$$

Using the sensitivity lower bounds $\mu(p) \gtrsim \Delta_p/\text{cost}(P, A)$ and $\mu(p) \gtrsim \text{cost}(p, A)/\text{cost}(P, A)$, together with $\text{cost}(P, A) \lesssim \text{cost}(P, S)$, gives $|X_i| \lesssim (\varepsilon m)^{-1}$. Therefore $|X_i - \mathbb{E} X_i| \lesssim (\varepsilon m)^{-1}$ as well. $\qquad\square$

Now we finish the proof of the claim.

*Proof of Claim A.7.* By Claim A.10, Bernstein's inequality gives

$$\Pr\left[\left|\sum_{i=1}^{m}(X_i - \mathbb{E} X_i)\right| > \varepsilon/4\right] \leq \exp\left(-\Omega(\varepsilon^2 m)\right).$$

By the choice of

$$m = C\varepsilon^{-2}\left(k\log(k/\delta) + \log N\right),$$

and for a sufficiently large constant $C$, this failure probability is at most $\delta/(4N)$ for any fixed $S \in \mathcal{S}$. Taking a union bound over all $N$ solutions in $\mathcal{S}$ gives, with probability at least $1 - \delta/4$,

$$\sup_{S \in \mathcal{S}}\left|\frac{A(S) - C(S, \Omega)}{\text{cost}(P, S)}\right| \leq \varepsilon/4. \qquad\square$$

## A.4. Event $E$ Occurs With High Probability

We prove Claim A.2. Fix a cluster $C_j$. We prove the two conditions in Event $E$ separately and then union bound over all clusters.

**Cluster weight.** Define
$$Y_i := w(q_i)\mathbf{1}[q_i \in C_j].$$

Then $\sum_i Y_i$ is the total coreset weight in $C_j$, and $\mathbb{E}[\sum_i Y_i] = |C_j|$. Since $\mu(p) \gtrsim 1/(k|C_j|)$ for $p \in C_j$, we have

$$0 \leq Y_i \lesssim \frac{k|C_j|}{m}, \qquad \sum_{i=1}^{m}\text{Var}(Y_i) \lesssim \frac{k|C_j|^2}{m}.$$

Therefore, Bernstein's inequality implies

$$\Pr\left[\left|\sum_i Y_i - |C_j|\right| > \varepsilon|C_j|\right] \leq 2\exp\left(-\Omega(\varepsilon^2 m/k)\right).$$

**Cluster $A$-cost.** Define
$$Z_i := w(q_i)\text{cost}(q_i, A)\mathbf{1}[q_i \in C_j].$$

Then $\sum_i Z_i$ is the sampled $A$-cost in $C_j$, and $\mathbb{E}[\sum_i Z_i] = \text{cost}(C_j, A)$. Assuming $\text{cost}(C_j, A) > 0$, the sensitivity lower bound $\mu(p) \gtrsim \text{cost}(p, A)/(k\text{cost}(C_j, A))$ gives

$$0 \leq Z_i \lesssim \frac{k\text{cost}(C_j, A)}{m}, \qquad \sum_{i=1}^{m}\text{Var}(Z_i) \lesssim \frac{k\text{cost}(C_j, A)^2}{m}.$$

Therefore,

$$\Pr\left[\left|\sum_i Z_i - \text{cost}(C_j, A)\right| > \varepsilon\text{cost}(C_j, A)\right] \leq 2\exp\left(-\Omega(\varepsilon^2 m/k)\right).$$

If $\text{cost}(C_j, A) = 0$, then all points in $C_j$ have zero $A$-cost, so the second condition holds trivially.

**Union bound.** For a sufficiently large constant in the sample size, each failure probability is at most $\delta/(8k)$. A union bound over the two conditions and all $k$ clusters gives $\Pr[E] \geq 1 - \delta/4$.

### A.5. Cost of Far Clusters is Preserved

We prove Lemma A.5. By Claim A.2, Event $E$ holds with probability at least $1 - \delta/4$. We condition on Event $E$ and prove the desired bound deterministically.

Fix $S \in \mathcal{S}$. For a cluster $C_j$ that is far from $S$, write $A_j := \text{cost}(a_j, S)$. The key point is that $|C_j|A_j$ is a good proxy for the true cost of $C_j$, while the corresponding sampled quantity is controlled by Event $E$.

**Claim A.11.** *For every cluster $C_j$ that is far from $S$,*

$$|\text{cost}(C_j, S) - \text{cost}_\Omega(C_j, S)| \lesssim \varepsilon|C_j|A_j.$$

*Proof.* By Claim A.9(i), for every $p \in C_j$,

$$|\text{cost}(p, S) - A_j| \lesssim \sqrt{A_j\text{cost}(p, A)} + \text{cost}(p, A).$$

Summing over $p \in C_j$ and applying Cauchy–Schwarz gives

$$|\text{cost}(C_j, S) - |C_j|A_j| \lesssim |C_j|\left(\sqrt{A_j\Delta_j} + \Delta_j\right).$$

Since $C_j$ is far from $S$, we have $\Delta_j \leq \varepsilon^2 A_j$, and therefore

$$|\text{cost}(C_j, S) - |C_j|A_j| \lesssim \varepsilon|C_j|A_j.$$

Now let

$$W_j := \sum_{q \in C_j \cap \Omega} w(q), \qquad \Phi_j := \frac{1}{W_j}\sum_{q \in C_j \cap \Omega} w(q)\text{cost}(q, A).$$

Applying the same argument to the weighted sampled points gives

$$|\text{cost}_\Omega(C_j, S) - W_jA_j| \lesssim W_j\left(\sqrt{A_j\Phi_j} + \Phi_j\right).$$

By Event $E$, $W_j = (1 \pm \varepsilon)|C_j|$ and $\Phi_j = (1 \pm O(\varepsilon))\Delta_j$. Since $\Delta_j \leq \varepsilon^2 A_j$, this implies

$$|\text{cost}_\Omega(C_j, S) - W_jA_j| \lesssim \varepsilon|C_j|A_j.$$

Event $E$ also gives

$$|W_jA_j - |C_j|A_j| \leq \varepsilon|C_j|A_j.$$

The claim follows by the triangle inequality. $\square$

Summing Claim A.11 over all clusters far from $S$ gives

$$|\text{cost}(P_F(S), S) - \text{cost}_\Omega(P_F(S), S)| \lesssim \varepsilon \sum_{j:\, C_j \text{ far from } S} |C_j|\text{cost}(a_j, S) \leq \varepsilon\sum_{j=1}^{k} |C_j|\text{cost}(a_j, S).$$

By Claim A.3, the last quantity is at most $O(\varepsilon)\text{cost}(P, S)$. Rescaling constants gives Lemma A.5.

### A.6. A note on composability

Composability refers to the property that the union of strong coresets for disjoint data sets are a strong coreset for the union. This property follows immediately from the coreset definition and the $k$-means cost function.

Approximation preserving coresets are not composable by definition. Nevertheless, since we are obtaining them via sensitivity sampling, we can argue composability. To begin, we notice that oversampling yields strong coresets, which are composable and which we can then downsample to obtain an APC. If the oversampling is too expensive, as might the case in finite metrics, then we observe that the number of solutions induced by merging $t$ APCs each of size at most $m$, increases from $\binom{m}{|S|}$, where $|S|$ is the size of the witness sets to $t \cdot \binom{m}{|S|}$. Thus, the oversampling only increases by a factor $\log t$.

# B. Further Experimental Evaluation

First, we present detailed cost-ratio plots across different values of $k$ for each dataset. Our plots indicate that, across all three algorithms, the cost-ratio curve as a function of $m/k$ is essentially invariant to the choice of $k$. Consequently, we aggregate results by fixing $m/k$ and averaging the cost ratio over all tested values of $k$ and corresponding $m$, and study the resulting curve as a function of $m/k$.

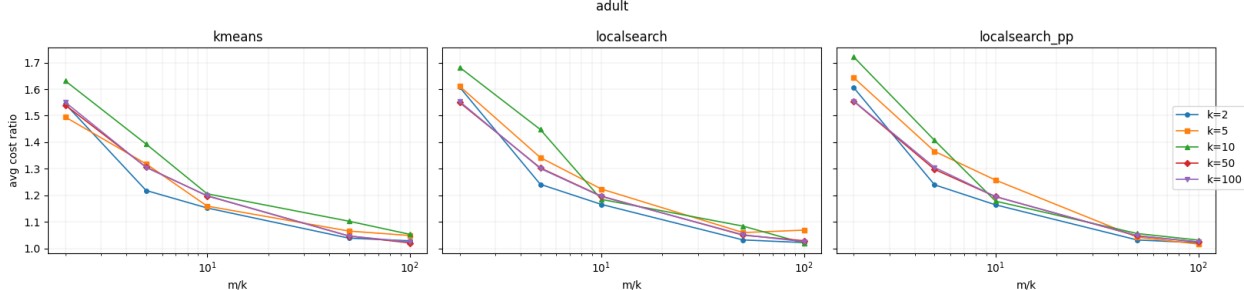

*Figure 3.* Adult dataset

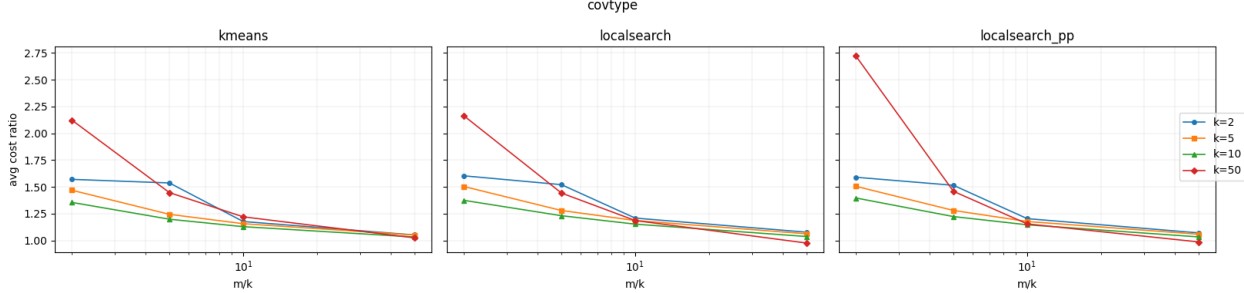

*Figure 4.* Covertype dataset

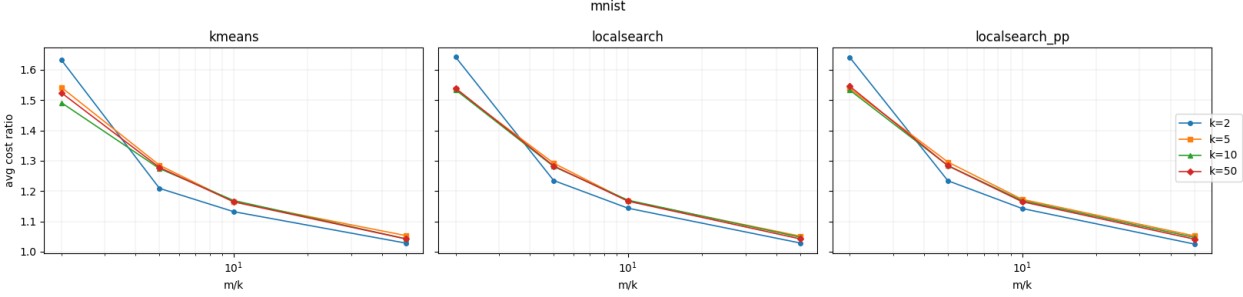

*Figure 5.* MNIST dataset

Second, we present aggregate plots for MNIST and Covertype (Figure 6), obtained by averaging the cost ratio over all runs with the same $m/k$.

Finally, we report tables (Table 1, Table 2, Table 3), of solution costs obtained by running the three algorithms on the full datasets. This is included to illustrate the universality of our APCs: the guarantees hold across algorithms, even when their baseline solution qualities differ.

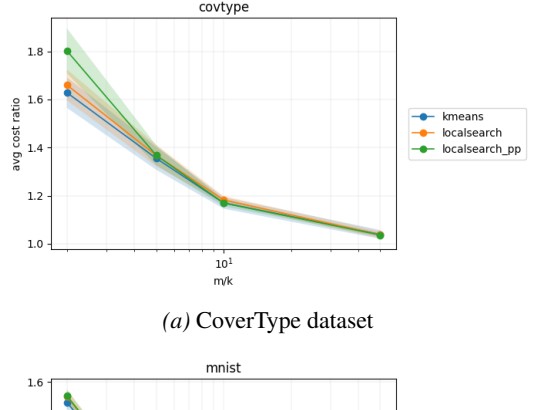

*(a)* CoverType dataset

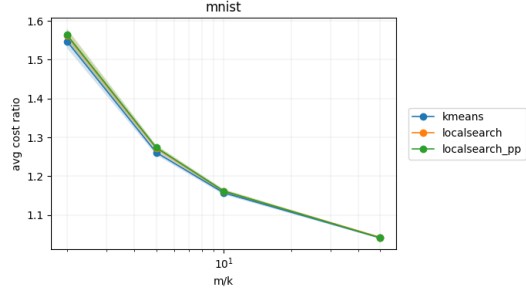

*(b)* MNIST dataset

*Figure 6.* Average cost ratio vs. $m/k$ ( MNIST, Covertype) for each algorithm: costs are evaluated on the full dataset and averaged over all $k$ and trials.

| $(k, m)$ | K-MEANS++ | LOCAL-SEARCH | LOCAL-SEARCH++ |
|---|---|---|---|
| (2,10) | 5.646e+05 | 5.749e+05 | 5.749e+05 |
| (5,50) | 4.384e+05 | 4.454e+05 | 4.430e+05 |
| (10,100) | 3.536e+05 | 3.438e+05 | 3.360e+05 |
| (50,500) | 2.477e+05 | 2.459e+05 | 2.453e+05 |

*Table 1.* Cost on the scaled dataset (objective evaluated on the full dataset).

| $(k, m)$ | K-MEANS++ | LOCAL-SEARCH | LOCAL-SEARCH++ |
|---|---|---|---|
| (2,10) | 4.183e+06 | 4.281e+06 | 4.282e+06 |
| (5,50) | 3.561e+06 | 3.578e+06 | 3.587e+06 |
| (10,100) | 3.236e+06 | 3.231e+06 | 3.220e+06 |
| (50,500) | 2.530e+06 | 2.527e+06 | 2.522e+06 |

*Table 2.* Cost on the scaled dataset (objective evaluated on the full dataset).

| $(k, m)$ | K-MEANS++ | LOCAL-SEARCH | LOCAL-SEARCH++ |
|---|---|---|---|
| (2,10) | 4.638e+07 | 4.495e+07 | 4.495e+07 |
| (5,50) | 3.181e+07 | 3.205e+07 | 3.182e+07 |
| (10,100) | 2.723e+07 | 2.760e+07 | 2.747e+07 |
| (50,500) | 8.070e+06 | 7.750e+06 | 5.793e+06 |

*Table 3.* Cost on the scaled dataset (objective evaluated on the full dataset).

