# OpenReview forum: "Approximation Preserving Coresets"
_ICML.cc/2026/Conference — ICML 2026 regular_

### Official Review · Reviewer_jyV8 · 2026-03-09

**Soundness:** 3
**Presentation:** 3
**Significance:** 3
**Originality:** 3
**Overall Recommendation:** 4
**Confidence:** 1

**Summary:**

This paper introduces the novel concept of "approximation-preserving coresets" (APCs) for the k-means clustering problem. Unlike traditional strong coresets, which must preserve the cost of all possible solutions, an APC only guarantees that a good solution computed on the coreset can be efficiently post-processed into a good solution for the full dataset. This relaxation allows for the construction of significantly smaller summaries.

**Compliance With Llm Reviewing Policy:**

Affirmed.

**Final Justification:**

Thanks for the authors' rebuttal.

**Key Questions For Authors:**

I don't see any significant weaknesses in this paper. I'm unfamiliar with this area, and I don't know why ICML assigned this paper to me.

**Limitations:**

Yes

**Strengths And Weaknesses:**

Strengths:

* The idea of relaxing the strong coreset guarantee to focus on the outputs of approximation algorithms is both novel and practically motivated. The paper clearly articulates why such a relaxation could lead to smaller summaries, directly addressing a key limitation of coreset theory.

* The paper is exceptionally well-written. The technical overview in Section 1.3, introducing the concept of witness sets, provides an intuitive and accessible roadmap for the rest of the paper. The organization is logical, and the progression from Euclidean to general metrics is natural.

---

> ### Author Rebuttal · Authors · 2026-03-29
>
> We thank the reviewer for their comments on our submission.

---

> > ### Author Rebuttal · Reviewer_jyV8 · 2026-04-02
> >
> > Thanks for the authors' rebuttal.

---

### Official Review · Reviewer_cFay · 2026-03-12

**Soundness:** 4
**Presentation:** 3
**Significance:** 4
**Originality:** 3
**Overall Recommendation:** 5
**Confidence:** 5

**Summary:**

Coresets for k-means clustering have been widely studied in recent years. An $\epsilon$-strong coreset is a (weighted) subset of the dataset that preserves the cost function on arbitrary solutions. Almost tight size bounds for strong coresets have been established through a line of research.

In particular, for k-means clustering, the $k/\epsilon^4$ upper bound can not be improved in the general case. A natural question is whether a smaller size bound can be obtained if one relaxes the strong coreset requirement on all candidate solutions. A popular notion is called weak coreset, where the coreset only needs to preserve the objective value on near-optimal solutions. It was not clear whether there was a gap between the size of an $\epsilon$ weak coreset and an $\epsilon$ strong coreset.

This paper actually answers this question affirmatively by constructing an $\epsilon$-approximation preserving coreset (which is claimed to be stronger than a weak coreset) of size $\O(k/\epsilon^3)$ in Euclidean space. Such a coreset suffices to preserve the quality of an arbitrary approximation algorithm. Technically, the analysis of their sampling algorithms relies on a very simple observation that one only needs to preserve the candidate centers, which are the mean of $O(1/\epsilon)$ many data points. This idea is simple but very useful.

Authors also extend their results to a general metric space, but with a 4 approximation ratio loss. This result is less interesting. Moreover, authors have conducted experiments to show that one can use the new coresets to speed up existing clustering algorithms like local search. The experimental results are standard in coreset literature.

**Compliance With Llm Reviewing Policy:**

Affirmed.

**Final Justification:**

I have read the rebuttal and other reviews. I would like to keep my positive rating for this paper.

**Key Questions For Authors:**

1. It seems the techniques are ad hoc for k-means clustering. Can you obtain similar results for k-median? (Although for k-median, there already exists a strong coreset of size $\tilde{O}(k/\eps^3)$.

2. Is there a formal proof that APC is stronger than weak coreset? It does not seem immediate from the definition.

**Limitations:**

yes

**Strengths And Weaknesses:**

Strengths:
- The results separate the size bounds of strong coreset and weaker notions. To my understanding, this is the first such result.
- The idea is clever, which uses simple observations elegantly.
- The paper is mostly well-written and easy to follow.

Weaknesses:

- Some references are incomplete or duplicate.

---

> ### Author Rebuttal · Authors · 2026-03-29
>
> We would like to thank the reviewer for the useful comments, questions and suggestions.
>
> ---
>
> **Question.** It seems the techniques are ad hoc for k-means clustering. Can you obtain similar results for k-median? (Although for k-median, there already exists a strong coreset of size $\tilde O(k \varepsilon^{-3})$).
>
> **Response.** The reviewer's observation is indeed correct. For the $k$-median problem, we do not obtain smaller coresets if we want the Approximation Preserving Coresets guarantee (as compared to a strong coreset).
>
> ---
>
> **Question.**
>     Is there a formal proof that APC is stronger than weak coreset? It does not seem immediate from the definition.
>
> **Response.**
> One issue with this question is that weak coresets have been defined in many different ways in literature. The common feature of previous definitions is that they allow one to extract a $(1+\varepsilon)$-approximation. So in a sense, weak coresets are only guaranteed to preserve near optimal solutions, i.e., they only ensure that a (near) optimal solution computed on the weak coreset is also a  (near) optimal solution for the full dataset.
>
> For APC's however, we require that any $\alpha$-approximate solution can be post-processed to find a $\alpha \beta$ approximate solution for the full dataset. In terms of the definition (see the Technical Overview section), we require the mapping property (any good solution for the coreset can be mapped to a good solution for the full dataset) , in addition to preserving the cost of an optimal solution.
>
> An APC can be used in conjunction with any approximation algorithm, such as the popular and fast $k$-means++ or local search algorithms. By contrast, weak coresets require using a $(1+\varepsilon)$-approximation algorithm, all of which have running times $\exp(k/\varepsilon)$ or more.
> It is conceivable that weak coreset algorithms yield summaries with stronger properties, but it does not follow from the definition.

---

> > ### Author Rebuttal · Reviewer_cFay · 2026-04-02
> >
> > Thanks for the answers.

---

### Official Review · Reviewer_qE5q · 2026-03-13

**Soundness:** 3
**Presentation:** 3
**Significance:** 3
**Originality:** 3
**Overall Recommendation:** 4
**Confidence:** 3

**Summary:**

The paper presents a coreset technique for approximate k-means algorithms. More specifically, previous works constructed coresets that allowed for approximating the k-means cost function on any set of centers. In contrast, this paper proposes to compute a coreset that approximates the k-means cost function on the centers computed by approximate k-means algorithms (i.e., on a reduced set of centers). Let A be an approximation algorithm for k-means. The idea leverages Sensitivity  Sampling [Bansal et al 2024]: a coreset is computed with Sensitivity Sampling, then a procedure  improves the solution computed by A on the coreset to reach a better quality.

**Compliance With Llm Reviewing Policy:**

Affirmed.

**Final Justification:**

I thank the authors for the feedback. I'm towards a weak accept.

**Key Questions For Authors:**

- Is the size of strong coresets fixed (e.g. Bansal's one) for fixed input size or does it depend on the actual input? What about your solution?
- Can be any strong coreset (without details on the construction procedure) be "simplified" to be APC?

**Limitations:**

yes

**Strengths And Weaknesses:**

Strengths:
- I like the idea to obtain a smaller coresets by exploiting the fact that coresets are used with approximate solutions.
- The results are technical sounding, and the introduction explains the main results of the paper.

Weaknesses:
-The experimental evaluation is quite weak: the paper analyzes the quality of clustering algorithms using their coresets; however there is no experimental comparison with other coresets technique (i.e., with strong coresets). Moreover the provided Colab project only contains a readme file.


----
I apologize for my initial review, which was based on a misunderstanding of the results in the paper.

---

> ### Author Rebuttal · Authors · 2026-03-29
>
> We thank the reviewer for their useful feedback, comments, and questions.  We note that the Open4Science link included with the submission did not reflect some later updates, due to a synchronization issue affecting the linked materials. We apologize for this inconvenience. The link now contains all the relevant code.
>
> ---
>
> **Question:** Is the size of strong coresets fixed (e.g. Bansal's one) for fixed input size or does it depend on the actual input? What about your solution?
>
> **Response:** Whether the coreset size depends on the input size depends on the metric space in which the points lie.
>
> For Euclidean space, both strong coresets and APCs have sizes independent of the number of points $n$.  The strong coreset construction for Euclidean spaces from Bansal et al. has size $\tilde{O}(\min(k\varepsilon^{-4}, k^{1.5} \varepsilon^{-2}))$ where  $k$ denotes the number of clusters and $\varepsilon$ is the accuracy parameter. Our APC for Euclidean space has better guarantees; it is of size $\tilde{O}(k\varepsilon^{-3})$.
>
> For finite metric spaces, it is known from previous work that there are metric spaces where any $(1+\varepsilon)$-strong coreset has size $\Omega(k \varepsilon^{-2} \log n)$. In our paper we show that such bad metric spaces  also exist for approximation preserving coresets: in particular we show that there are metric spaces where any $\beta$-APC for $\beta < 4$ must have size $\Omega(\sqrt{\log n})$.
>
> ---
> **Question:** Can  any strong coreset (without details on the construction procedure) be "simplified" to be APC?
>
> **Response:** Yes, all strong coresets are $(1+\varepsilon)$-APCs as they preserve the cost of all solutions. So given a strong coreset one can recover an $\alpha(1+\varepsilon)$-approximate  $k$-means solution by finding an $\alpha$-approximate solution for the strong coreset. It is not clear however, that a space savings can be achieved beyond that. Verifying both the strong coreset as well as the APC property seems to be a hard problem. Naturally these problems are in coNP, for which even heuristic algorithms are difficult to find.

---

> > ### Author Rebuttal · Reviewer_qE5q · 2026-04-02
> >
> > I thank the authors for the feedback.
> > For the last point I meant: is it possible to post process a strong coreset by reducing its size and weak its property to approximate preserving coreset?

---

> > > ### Author Response · Authors · 2026-04-02
> > >
> > > One could always run our algorithm. If the input is guaranteed to be a strong coreset, then our algorithm likely will reduce the size, as the worst case upper bound on the size of the APC it produces is smaller than the worst case lower bound for any strong coreset. Any approximation preserving coreset of a strong coreset is also an approximation preserving coreset of the original point set and the algorithm could use the coreset instead of the original point set as input, without any issues.
> > >
> > > But if the question is whether every strong coreset can be reduced in size if we are only aiming to find an approximation preserving coreset, then it easy to find instances where this cannot be the case. As a very simple example: If the instance only consists of $k$ distinct points, the point set itself is a strong coreset of size $k$ and removing any one of these points will neither be an approximation preserving coreset, nor a strong coreset.

---

### Official Review · Reviewer_SoY1 · 2026-03-20

**Soundness:** 3
**Presentation:** 3
**Significance:** 3
**Originality:** 3
**Overall Recommendation:** 5
**Confidence:** 3

**Summary:**

The paper introduces the notion of approximation preserving coreset for the k-means clustering problem, which can be seen as something in-between a weak coreset and a strong coreset. Strong coreset preserves the cost of all feasible solutions, weak coreset only preserves the cost of the optimal solution, but an approximation scheme on a weak coreset may not give a good approximate solution for the whole input. APCs aim to preserve the performance of approximation algorithms rather than the cost of all possible solutions. The main theoretical result shows that this relaxation leads to somewhat smaller coresets. In Euclidean space, the authors construct a (1+\eps)-APC of size $\approx k\epsilon ^{-3}$, improving over the larger worst-case bounds required for strong coresets. In arbitrary metric spaces, they obtain a (4+\eps)-APC of size O(k\epsilon^{-2}), notably independent of the input size n, and show that achieving a factor better than 4 necessarily requires larger coresets that depend on n.


In addition to these structural results, the paper shows that APCs yield efficient algorithmic consequences. In particular, they imply a fast fixed-parameter tractable (FPT) approximation algorithm for metric k-means, with near-linear preprocessing time and exponential dependence only on k. The coresets are also composable and can be constructed efficiently via sensitivity sampling. Overall, the techniques seem to have sufficient novelty. Empirically, the authors demonstrate that relatively small coresets (on the order of O(k) points) suffice to retain the performance of standard approximation algorithms such as k-means++ and local search, providing evidence that APCs better capture practical behavior than traditional strong coresets.

**Compliance With Llm Reviewing Policy:**

Affirmed.

**Key Questions For Authors:**

NA

**Limitations:**

yes

**Strengths And Weaknesses:**

The main conceptual contribution of this paper is quite strong. The authors show how APCs of smaller size (than that of strong coresets) can be used to obtain reasonable approximations. On the technical side, the use of sensitivity sampling seems standard within many coreset based algorithms. The authors contribute a new concentration bound for sensitivity sampling that appears novel, although I'm not quite well versed with the recent literature on this to judge this novelty. The authors use experiments to support their strong theoretical contributions.

My only criticism is that it took me a few reads to appreciate the need for APCs, but I think I understand it now. Overall, this is a strong paper.

---

> ### Author Rebuttal · Authors · 2026-03-29
>
> We would like to thank the reviewer for their valuable feedback and for their positive reception of our results.
>
>
> We will include this discussion in the next version of the paper so as to clarify the motivation behind defining APCs and how they relate to strong and weak coresets.
> Intuitively, coresets in practice are constructed so that good approximation algorithms can be run on the coreset to recover a good clustering of the originally large dataset.
>
> At a high level,
> - Strong coresets preserve the approximation guarantee for all sets of centers.
> - Weak coresets preserve the approximation guarantee only for near-optimal centers.
>
> $\beta$-APCs capture an intermediate notion, motivated by practical applications: given any
> $\alpha$-approximate solution for the coreset, an $O(\alpha \beta)$-approximate  solution for the original dataset can be recovered.

---

> > ### Author Rebuttal · Reviewer_SoY1 · 2026-04-02
> >
> > Thank you for your comment.

---

### Decision · Program_Chairs · 2026-04-30

**Decision:**

Accept (regular)

**Comment:**

The paper introduces the notion of approximation-preserving coresets (APCs), which relax the standard strong coreset guarantee while maintaining the performance of approximation algorithms. Reviewers generally found this conceptual contribution both novel and practically motivated. The main strengths lie in establishing a clear separation between strong coresets and weaker notions, deriving improved coreset size bounds, and providing both theoretical justification and empirical support. Initial concerns focused on the clarity of motivation, comparisons to prior work, and the experimental evaluation, but the authors' rebuttal addressed these points satisfactorily by clarifying the conceptual positioning of APCs. While some reviewers noted that parts of the technique built on standard sensitivity sampling and that the experimental evaluation could be stronger, these were viewed as relatively minor issues.